# Q-Ground: Image Quality Grounding with Large Multi-modality Models

## ABSTRACT

Recent advances of large multi-modality models (LMM) have greatly improved the ability of image quality assessment (IQA) method to evaluate and explain the quality of visual content. However, these advancements are mostly focused on overall quality assessment, and the detailed examination of local quality, which is crucial for comprehensive visual understanding, is still largely unexplored. In this work, we introduce **Q-Ground**, the first framework aimed at tackling fine-scale visual quality grounding by combining large multi-modality models with detailed visual quality analysis. Central to our contribution is the introduction of the **QGround-100K** dataset, a novel resource containing 100k triplets of *(image, quality text, distortion segmentation)* to facilitate deep investigations into visual quality. The dataset comprises two parts: one with human-labeled annotations for accurate quality assessment, and another labeled automatically by LMMs such as GPT4V, which helps improve the robustness of model training while also reducing the costs of data collection. With the **QGround-100K** dataset, we propose a LMM-based method equipped with multi-scale feature learning to learn models capable of performing both image quality answering and distortion segmentation based on text prompts. This dual-capability approach not only refines the model's understanding of region-aware image quality but also enables it to interactively respond to complex, text-based queries about image quality and specific distortions. **Q-Ground** takes a step towards sophisticated visual quality analysis in a finer scale, establishing a new benchmark for future research in the area. Codes and dataset will be made available.

## CCS CONCEPTS

• **Computing methodologies** → Computer vision tasks.

## KEYWORDS

Image Quality Assessment, Large Multi-modality Models, Image Quality Grounding

## 1 INTRODUCTION

As a cornerstone in the domain of digital imaging, Image Quality Assessment (IQA) covers a wide range of methods designed to evaluate the perceptual quality of visual content similarly to human visual system (HVS). With the rapid increase of digital content, IQA

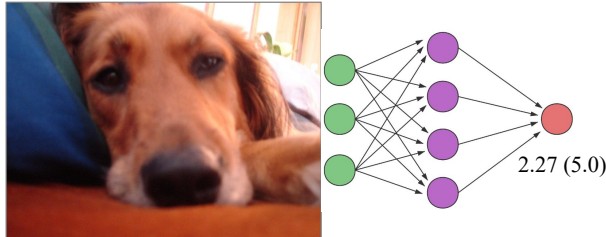

(a) Visual quality scoring

*User:* Please evaluate the quality of the image.

- - - - - - - - - - - - - - - - - - - - - - - - - - - - - - - - -

*Assistant:* The overall clarity of this image is very low, with weak lighting. The main subject, a dog, has lost most of its texture details and appears blurry. The background is also blurry. The composition is poor, with a slight tilt. The background is also blurry. The image has lost a significant amount of detail.

(b) Visual quality reasoning

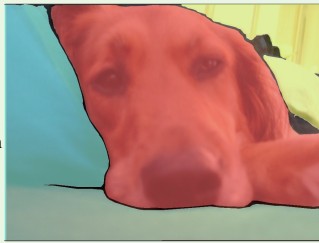

**(Continued)**
*User:* Please give distortion segmentation masks.

*Assistant:* The distortion segmentation masks of **blur**, **overexposure** and **noise** are [Seg] [Seg] [Seg].

(c) Visual quality grounding

**Figure 1: An example comparison between different tasks illustrates: (a) Visual quality scoring only provides a numerical score without an underlying rationale; (b) LMM-based reasoning offers clear explanations but lacks pixel-level comprehension; (c) the suggested approach to visual quality understanding not only facilitates quality reasoning but also delivers corresponding pixel-level distortion segmentation masks.**

is becoming more and more important in many areas, such as media streaming, user-generated photos and videos, smart-phone cameras and the growing field of AI-generated content. These various applications call for more powerful and understandable IQA methods to help create visual content with better quality and improve the experience of users.

Existing IQA methods has aimed to replicate the HVS's capability to distinguish and assess visual information, typically by correlating the mean opinion scores (MOS) labeled by humans with features derived from images. The performance of these methods has significantly improved with the advent of more powerful feature extractors, moving from hand-crafted features in traditional approaches [28, 29, 31, 68] to advanced deep neural networks [4, 15, 71]. Nonetheless, these works only give quality scores as results and face challenges in accurately evaluating and explaining

the details of image quality, particularly when it comes to local distortions and fine-grained analysis. Recent advances in Large Multi-Modality Models (LMMs) mark a new chapter for IQA, offering promising avenues for enhancing both the evaluation capabilities and the explanatory ability of IQA systems. For example, CLIPIQA [47] illustrates the zero-shot capabilities of multi-modality models in IQA, and Q-Bench [52] demonstrates near-human performance of GPT4V [33] in certain specific areas. However, despite these advancements, the application of LMMs in IQA remains focused on overall quality assessment. This narrow focus limits their utility for comprehensive visual analysis, particularly in contexts where fine-scale quality grounding and detailed understanding of local distortions are imperative.

In response to these challenges, we introduce the visual quality grounding task to the field of IQA for the first time, with the goal of bridging the gap in detailed image quality perception. As illustrated in Fig. 1, traditional methods of quality scoring yield a single numerical score without explanation, and existing quality reasoning methods does not account for local distortions. Our novel visual quality grounding strategy integrates pixel-level distortion segmentation with textual queries, substantially improving the fine-scale capabilities of IQA. The major problem in realizing this advancement is the lack of suitable datasets. Unlike standard segmentation tasks, the boundaries of distortion regions may exhibit minor variations due to individual subjective judgments. Therefore, we deploy two auxiliary methods to support the creation of more dependable mask annotations: 1) Preliminary segmentation of images using Semantic-SAM [22] to pinpoint potential areas of distortion; 2) Provision of a textual quality evaluation message during the annotation process to serve as a reference. Our dataset is constructed on top of Q-Instruct [53], which provides detailed textual explanations regarding the image quality. Consequently, we have compiled a visual quality grounding dataset containing 50K human-annotated triplet samples *(image, quality text, distortion segmentation)*. Recognizing the time-consuming and costly nature of human annotation, we additionally automate part of the dataset creation using GPT4V [33] because of its superior performance in overall quality evaluation [52]. By employing the set-of-mark strategy [63], we manage to collect an additional 50K samples for our dataset. These automatically labeled data can be easily generated, significantly broadening the diversity of our dataset. These two parts form our final dataset, **QGround-100K** , the first visual quality grounding dataset for fine-scale IQA.

The QGround-100K dataset enables the training of a quality grounding model for IQA. Rather than constructing a traditional visual grounding model that relies on separate embeddings for text and image as inputs, our aim is to develop a more capable and flexible multi-modality model that incorporates both text and images as inputs and outputs, akin to recent LMMs [18, 26, 64, 75]. Different from these existing methods, which primarily address high-level concepts, the visual quality grounding task places a greater emphasis on low-level and mid-level details. Consequently, we introduce a multi-scale feature abstractor (MSFA) to get quality-aware visual embeddings before merging them with text embeddings into pre-trained large language model (LLM), thereby augmenting LMM's capacity for low-level perception. Furthermore, we train our model using a diverse dataset comprising high-level multi-modality data, the

quality reasoning dataset [53], and the newly proposed QGround-100K . These varied datasets enable our model to undertake complex tasks, such as answering text-based questions about image content and quality, as well as conducting distortion segmentation. By integrating these features, our approach smoothly combines fine-scale and overall quality perception capabilities within the interactive analysis of visual contents, setting a new benchmark for future explorations in the field.

Our contributions can be summarized as follows:
- To the best of our knowledge, we are the first to present framework aimed at fine-scale visual quality grounding, using the strengths of LMMs for detailed visual quality analysis.
- We construct the QGround-100K dataset, the first-of-its-kind dataset comprising 100K samples designed to support deep investigations into visual quality, encompassing both human-labeled and LMM-generated annotations.
- We introduce multi-scale visual feature abstractor for LMM-based visual quality grounding. The model is capable of performing image quality assessment and distortion segmentation with textual prompts, thus facilitating a fine-scale understanding of quality and interactive engagement with visual content.
- Our work establishes a new benchmark for future research in IQA, paving the way for more sophisticated and fine-grained analyses of image quality.

## 2 RELATED WORKS

### 2.1 Image Quality Assessment

*2.1.1 Previous Methods.* Current methods in IQA can be broadly divided into Full-Reference (FR) and No-Reference (NR) techniques. FR methods assess the discrepancy between a reference image and its distorted counterpart. The widely recognized Peak Signal-to-Noise Ratio (PSNR) evaluates this difference on a pixel-wise basis, whereas the Structural Similarity Index (SSIM) [49, 50] enhances this evaluation by incorporating structural similarity features, thereby inspiring several subsequent studies [19, 20, 40, 41, 62, 67, 70]. Learning-based approaches [2, 4, 7, 17, 36, 71] have come to dominate FR IQA with significantly better performance, providing more accurate and reliable assessments of image quality. However, the necessity for a reference image limits their applications.

The development of NR-IQA which is more challenging has followed a similar trajectory to that of FR-IQA. Traditional methods, exemplified by NIQE [31], rely on natural scene statistics [1, 23, 28, 30, 32, 69]. In contrast, recent advancements [15, 43, 44, 51, 60, 73] in deep learning enable methods to directly learn to estimate MOS in an end-to-end fashion. The efficacy of these deep learning models is closely related to the datasets they are trained on, resulting in capabilities that are less interpretable. For instance, a model trained on an aesthetic assessment dataset may excel at evaluating aesthetic quality [12, 45], yet recognizing this specialty from its output scores is not straightforward. The emergence of multi-modality models, notably CLIP [38], has inspired recent initiatives [13, 16, 47, 74] to integrate the descriptive power of textual information with IQA, proving beneficial. Consequently, the latest works [14, 53–56, 66] employ LMMs in IQA, significantly enhancing both performance and interpretability, and leading to a new era in IQA research.

**Table 1: Comparison of existing public IQA datasets and the proposed QGround-100K.**

| Type | Dataset | MOS | Text | Seg |
|------|---------|-----|------|-----|
| FR | Traditional datasets [20] [42] [35] [24] [71] | ✔ | ✘ | ✘ |
| NR | Traditional datasets [9] [11] [8] [10] [65] | ✔ | ✘ | ✘ |
| | Q-Instruct [53] | ✔ | ✔ | ✘ |
| | **QGround-100K** | ✔ | ✔ | ✔ |

Despite these significant advancements, current IQA methods are limited to providing either a global score or a textual evaluation and lack the capability to evaluate image quality within the context of local distortions. Our work aims to bridge this gap.

*2.1.2 Existing Datasets.* There are numerous datasets that have been pivotal in the development of both FR and NR IQA algorithms, as summarized in Tab. 1. The FR datasets typically include images with synthetic distortions like Gaussian blur and white noise, where subjects compare two images and assign a quality score, which is a process that can introduce score ambiguities. To address this, BAPPS [71] introduces a two-alternative forced choice to reduce score uncertainty. Traditional NR datasets typically require subjects to provide a simple score. SPAQ [8] further requires quality ratings specific to various distortions and contents. While these datasets are invaluable for training and benchmarking IQA models, the reliance on simple quality scores limits their interpretability. Therefore, Q-Instruct [53] introduces textual quality descriptions, significantly enhancing the interpretability of IQA datasets. Nevertheless, these datasets mainly focus on global quality assessments, paying less attention to local distortion identification and detailed quality analysis. This limitation narrows their utility in applications demanding precise local distortion analysis, such as in image enhancement and editing tasks. The proposed QGround-100K dataset seeks to bridge this gap with comprehensive annotations including MOS, textual evaluations, and segmentation masks, establishing a more versatile tool for advanced IQA applications.

## 2.2 Visual Grounding with LMM

Visual quality grounding has long been an important task in computer vision, serving as a bridge between visual data and textual descriptions. Prior visual grounding, also known as referred expression comprehension, is mostly like a text conditioned localization task, see [37] for a comprehensive survey. The evolution of LMMs has significantly influenced recent developments. Innovations such as Kosmos-2 [34], Shikra [5], GPT4RoI [72], VisionLLM [48] *etc.*, have successfully merged generative LMMs with localization tasks, facilitating human-model interactions at the region level. Recent advancements, notably LISA [18], GLaMM [39], and PixelLM [75], have significantly improved upon existing methods by introducing pixel-level segmentation. However, the application of LMMs in the specific context of image quality assessment and visual grounding remains relatively unexplored. Our work takes a pioneering step forward in advancing fine-grained quality perception, marking a notable contribution to this evolving landscape.

**Table 2: The image sources and statistics of QGround-100K .**

| Image Sources | Original | Human labeled (Q-Pathway) | GPT4V-labeled |
|---------------|----------|---------------------------|---------------|
| KonIQ-10K [11] | 10,373 | 5,182 | 5,168 |
| SPAQ [8] | 11,125 | 10,797 | – |
| LIVE-FB [65] | 39,810 | 800 | 38,946 |
| LIVE-itw [9] | 1,169 | 200 | 969 |
| AGIQA-3K [21] | 2,982 | 400 | 2,568 |
| ImageRewardDB [59] | 50,000 | 584 | 2,947 |
| # Image | – | 17,963 | 50,599 |
| # Annotation | – | 52,924 | 50,599 |

## 3 THE QGROUND-100K DATASET

In this section, we provide details about the process of constructing the QGround-100K dataset, which lays the foundation for enabling visual quality grounding. We discuss the sources of our data in Sec. 3.1, and outline the annotation pipeline involving both human annotators and GPT4V in Sec. 3.2. Additionally, we provide an analysis and statistics of labels obtained from human annotators and GPT4V in Sec. 3.3, offering insight into the dataset's composition and the reliability of its annotations.

### 3.1 Data Collection

To develop a model capable of visual quality grounding, a dataset comprising triplet samples is essential: an input image, associated quality descriptive text, and ground truth distortion segmentation masks. Since the Q-Instruct [53] dataset provides comprehensive text descriptions for images from diverse resources, we choose to build QGround-100K upon it, as summarized in Tab. 2. We exclude 1K synthetic distorted images from COCO due to their focus on global distortions. As outlined in Tab. 2, for images within the Q-Pathway that already have human-labeled texts, we complement them with human labeled segmentation masks. To enrich the diversity of images, we include the rest images from IQA datasets for GPT4V labeling. Recognizing the rising popularity of AI-generated images, we also add 5.5K images from [21, 59]. The accompanying quality text is generated using latest Co-Instruct model[1], chosen for its performance comparable to that of GPT4V.

### 3.2 Data Annotation

To streamline the annotation process, we have chosen five prevalent types of distortions for mask annotation: blur, overexposure, noise, jitter and low light. These categories were selected for their frequency and significance in impacting visual quality across a wide range of images according to the report in Q-Instruct [53]. Figure 2 showcases the comprehensive data annotation pipeline, which incorporates both human and GPT4V annotation stages. Below, we provide detailed explanations for each phase within the pipeline, ensuring clarity and insight into our systematic approach for annotating the QGround-100K dataset.

*3.2.1 Human Annotation.* In the human annotation phase, 15 trained annotators with solid educational backgrounds are presented with (image, quality description) pairs. Their task is to segment out the

---

[1]https://huggingface.co/q-future/co-instruct

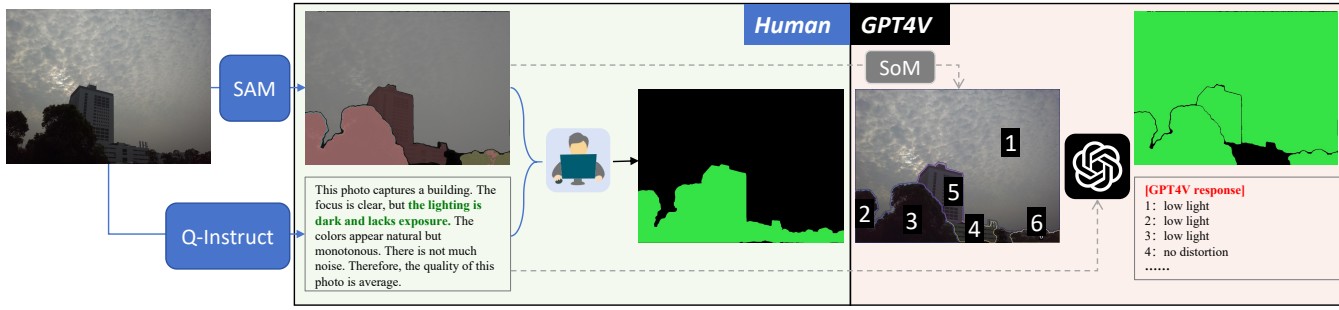

**Figure 2: The data annotation pipeline incorporates both human expertise and GPT-4V capabilities. Firstly, the input image undergoes pre-segmentation using SAM [22]. In the human annotation phase, subjectives need to identify and categorize types of distortions, with quality description texts from humans as reference. The subjective is free to adjust borders generated by SAM. In the GPT4V annotation phase, the reference for quality is generated by the Q-Instruct model. Then, each region is marked with a number, which is then coupled with the quality text and forwarded to the GPT4V model. Finally, the model outputs the types of distortions present in each specified region.**

distorted regions within the images and categorize the types of distortions present. To minimize ambiguity in the annotations, annotators are instructed to consult the provided quality descriptions throughout the annotation process. This step is crucial to ensure that the regions of interest related to quality assessment are accurately highlighted. Additionally, a pre-segmentation step utilizing SAM [22] is implemented to improve uniformity in the segmentation boundaries associated with specific distortions. Despite this automated assistance, annotators retain the judgement to manually adjust the boundaries. This flexibility acknowledges that SAM's segmentation may prioritize object areas over actual distortion locations, thereby allowing for more precise identification of quality-related distortions.

*3.2.2 GPT4V Annotation.* Following the human annotation phase, the GPT-4V annotation employs the Set-of-Mark (SoM) technique to facilitate mask annotation. As depicted in Fig. 2, the images are initially segmented using SAM and labeled with numbers. Subsequently, GPT4V is provided with the same (image, quality description) pairs that were utilized in the human annotation process. Leveraging its profound comprehension of both visual and textual content, the model identifies and labels regions of distortion. This methodology allows GPT4V to autonomously generate segmentation masks for distorted regions within an image, informed by the provided quality descriptions. This automated process not only speeds up the annotation effort but also provides a scalable way to enrich the dataset with diverse interpretations of image quality, bridging the gap between human efforts and AI efficiency.

## 3.3 Analysis of QGround-100K

As summarized in Tab. 2, the QGround-100K dataset comprises rich images and annotations for visual quality grounding from both human and GPT4V. Here, we delve into the statistics between human and GPT4V annotations.

Given that the range and types of distortions can be subjective and may vary among different annotators, assessing the reliability of human annotations is crucial. To this end, we examine the agreement between various human annotations within the Q-Pathway

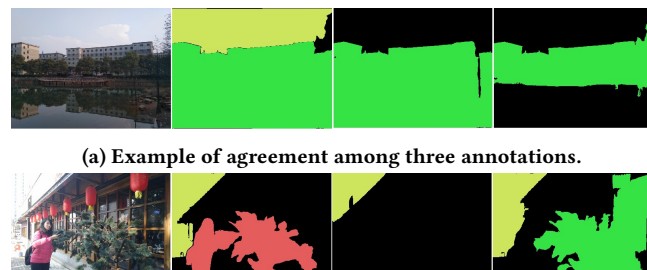

**(a) Example of agreement among three annotations.**

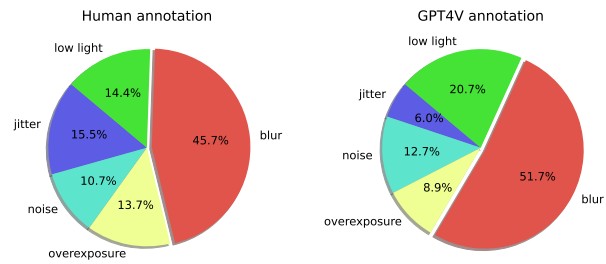

**(b) Example of disagreement among three annotations.**

| Dataset | KonIQ-10K | SPAQ | LIVE-FB | LIVE-itw | AGIQA-3K | ImageReward |
|---------|-----------|------|---------|----------|----------|-------------|
| Recall | 0.902 | 0.864 | 0.931 | 0.966 | 0.976 | 0.980 |

**(c) Pairwise recall between different annotators on different datasets.**

**Figure 3: Analysis of annotation agreement between different human subjectives.**

**Figure 4: Statistics of human and GPT4V parts separately.**

dataset, where each image is associated with at least three distinct quality text annotations. Different annotators label the same image but with different accompanying texts. As depicted in Fig. 3, we consider the results acceptable when one mask is a subset of another (Fig. 3(a)), and unacceptable when the same region is labeled with different types of distortions (Fig. 3(b)). To quantitatively evaluate the agreement score of human annotations, we employ the recall

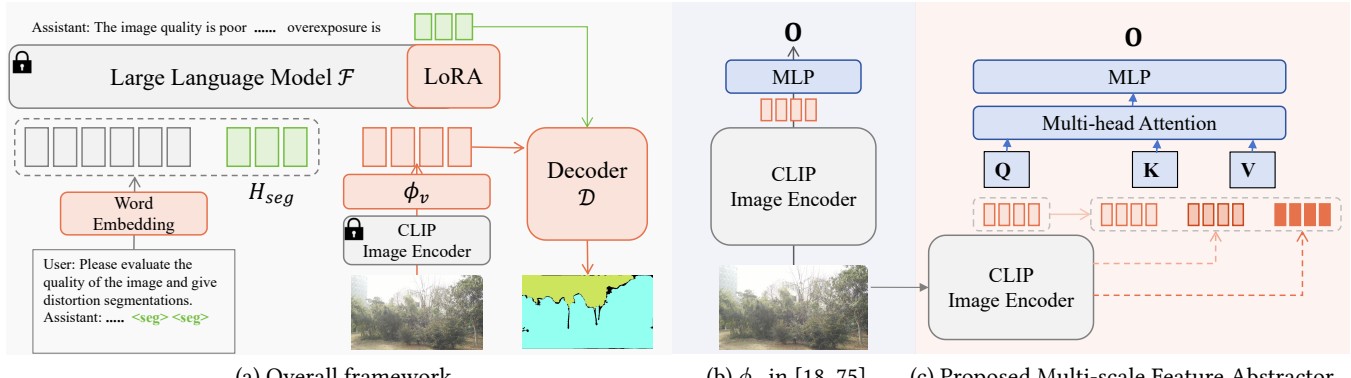

(a) Overall framework      (b) $\phi_v$ in [18, 75]      (c) Proposed Multi-scale Feature Abstractor

**Figure 5: The pipeline of our method. (a) The overall framework follows previous method [18, 75] and is designed to accept inputs of images and text, subsequently producing textual outputs and segmentation results. (b)(c): comparison of multi-modal projection block between previous works and our proposed multi-scale feature abstractor.**

of the pairwise intersection area over the smaller masks as follows:

$$\text{Recall} = \frac{1}{N}\sum_i^N \Big[\frac{1}{M_i}\sum_j^{M_i}\frac{A_j \cap B_j}{\min(A_j, B_j)}\Big], \qquad (1)$$

where $M_i = \binom{2}{m_i}$, $N$ is the number of images, $m_i$ is the number of masks for image $i$ and $A$, $B$ are the selected pairs. The findings, illustrated in Fig. 3(c), reveal that the agreement scores across different datasets are notably high, underscoring the reliability of our human annotation process. Regarding GPT4V annotations, it was observed that GPT4V consistently yields similar results across multiple runs when provided with appropriate prompts (detailed further in the supplementary materials). This analysis confirms the robustness and dependability of both human and GPT4V annotations within our dataset, laying a strong foundation for accurate visual quality grounding.

Besides, we provide an analysis of the distribution of distortion types found within both human and GPT4V annotations, as detailed in Fig. 4. Generally, the frequency of distortions observed roughly follows the order: blur > low light > overexposure ≈ noise. A notable deviation in this pattern is the higher incidence of jitter in human annotations compared to those by GPT4V. This difference likely comes from the Q-Pathway's substantial inclusion of images from SPAQ [8], a dataset composed of smartphone-captured images, which are prone to jitter due to hand movement. Conversely, the segment annotated by GPT4V primarily consists of web-crawled images, where jitter is less common. The similarity in the distribution of distortion annotations between human annotators and GPT4V highlights the GPT4V's effectiveness as a data generator, thereby validating its use in supplementing and expanding the dataset. Overall, the combined efforts of human and GPT4V annotations significantly enhance the diversity and utility of the dataset, providing a promising way to scale up datasets for visual quality grounding.

## 4 METHODOLOGY

Our objective is to develop a model capable of dialogues with users concerning the content and quality of images, while also executing

distortion region segmentation through text queries. Our framework follows the simple and efficient pipeline of PixelLM [75] (Sec. 4.1), and we improve the image to text projection block with multi-scale features to enhance quality-aware perception of the model (Sec. 4.2). Then, we train the model with multi-task datasets to enhance its capabilities (Sec. 4.3).

### 4.1 The Overall Framework

As illustrated in Fig. 5(a), the system processes both image inputs, denoted as $x_{img}$, and textual inputs, $x_{txt}$, to produce corresponding textual responses, $y_{txt}$, and segmentation masks, $y_m$. The inputs $I$ and $x_{txt}$ are firstly transformed into token embeddings, which are subsequently processed by a pre-trained large language model (LLM), such as LLaMA [46], to generate output tokens in an autoregressive manner. These tokens are then decoded to form $y_{txt}$. To facilitate the generation of segmentation outputs, we draw inspiration from previous works [18, 75] and introduce learnable segmentation tokens, represented as $H_{seg} = \{h^i \in \mathbb{R}^d\}|_{i=1}^N$, where $N$ represents the number of segmentation tokens and $d$ indicates the dimension of features. The segmentation masks, $y_m$, are generated using a decoder that takes the embeddings of $x_{img}$ as image input and $C_{seg}$ as condition inputs. This process involves the use of the pre-trained LLM, denoted as $\mathcal{F}$, and the CLIP image encoder, represented as $\mathcal{V}$. The overall pipeline of our framework is thus formulated as follows:

$$\{y_{txt}, H_{seg}\} = \mathcal{F}(\phi_v(\mathcal{V}(x_{img})), x_{txt}, H_{seg}), \qquad (2)$$

$$y_{seg} = \mathcal{D}(\mathcal{V}(x_{img}), H_{seg}), \qquad (3)$$

where $\mathcal{D}$ is a mask decoder same as [75], and $\phi_v$ is the projector from visual features to LLM embedding space.

As shown in Fig. 5(a), prior studies [18, 75] typically select straightforward Multilayer Perceptron (MLP) as $\phi_v$ and only use the final features from $\mathcal{V}(x_{img}) \in \mathbb{R}^{(h \times w) \times d_v}$, focusing mainly on high-level representations. Nonetheless, in our task centered on visual quality grounding, multi-scale features are critical for learning quality-associated perceptions, as evidenced by previous research [4]. Therefore, we introduce an innovative approach for $\phi_v$ that incorporates multi-scale features, as elaborated below.

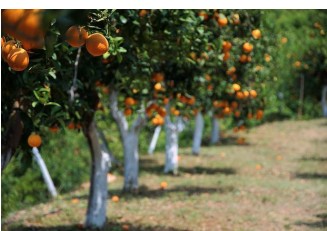
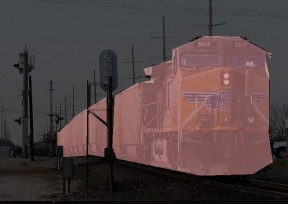
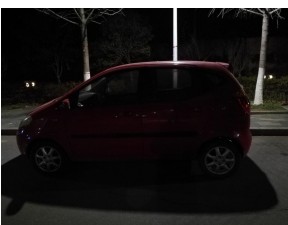
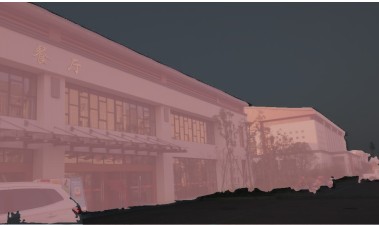

| *User:* What type of fruit trees are in the image? *Assistant:* The image features a row of orange trees. | *User:* What is train in this image? Please output segmentation mask. *Assistant:* Here is the segmentation result [SEG] | *User:* How does severe under exposure affect the image quality? *Assistant:* Loss of texture details and low sharpness. | *User:* Can you provide me with segment of the blur region? *Assistant:* Sure, blur mask is [SEG]. |

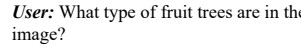

Visual question answering.      Semantic segmentation.      Visual quality reasoning.      Visual quality grounding.

**Figure 6: Example from various data sources for multi-task training of Q-Ground model.**

## 4.2 Multi-scale Feature Abstractor

The architecture of our proposed Multi-Scale Feature Abstractor (MSFA) is depicted in Fig. 5(c). Modern vision encoders mainly employ a vision transformer structure, exemplified by ViT/14 For an image with size $H \times W$, the feature dimensions from different layers remain the same as $\frac{H}{14} \times \frac{W}{14}$. A straightforward solution is to directly put multi-scale features into LLM, which will significantly increases computational cost due to the exponential rise in attention calculation as token length extends. For example, when $H = W = 448$ and 3 scales are used, the visual token length alone would be as long as $1024 \times 3$. On the other hand, such extensive visual tokens may not be essential owing to the redundancy in visual features. Recent study [57] shows that 256 tokens might be enough for integrating image features with LLM. Therefore, we present a multi-scale feature abstractor that employs a fixed-length query to distill useful information from multi-scale features. Given a set of multi-scale features $\mathbf{F} = \{f_i \in \mathbb{R}^{P \times d_v}\}$, where $f_i$ is the $i$-th layer feature from $\mathcal{V}(x_{img})$, the proposed MSFA can be calculated as

$$\mathbf{V} = \text{MHA}(\mathbf{Q}, \mathbf{F}, \mathbf{F}) \qquad (4)$$
$$\mathbf{O} = \sigma(\mathbf{V}W_1)W_2 \qquad (5)$$

where the MHA denotes multi-head attention, $\sigma$ is the activation function, $W_1, W_2$ are parameters of linear layers, and the query feature $\mathbf{Q} \in \mathbb{R}^{256 \times d_v}$. To simplify training, we use a pooled feature from the last layer of $\mathcal{V}(x_{img})$ as $\mathbf{Q}$, and $\mathbf{F}$ includes the last layer features in addition to features from several shallower layers.

## 4.3 Multi-task Training

To obtain a powerful LMM model which enables visual quality grounding into conversations with users, we employ various public data sources, as shown in Fig. 6.

To acquire a powerful LMM model capable of integrating visual quality grounding into interactive dialogues with users, we use a variety of publicly available data sources, as illustrated in Fig. 6. Our training dataset consists of four parts, detailed as below:

- *Visual question answering dataset.* This dataset enhances the model's understanding of visual content via question and answer pairs about the input image. We employ the LLaVA-Instruct-150K dataset [26] directly.
- *Semantic segmentation dataset.* A collection used to preserve the semantic segmentation ability of the model, avoiding model

overfitting to the distortion segmentation task. We include many different datasets for this part, *i.e.*, ADE20K [76], COCO images [25], COCO-stuff [3], as well as reasoning segmentation datasets from [18, 75].

- *Visual quality reasoning dataset.* The Q-Instruct dataset [53] is utilized to enable the model to answer questions regarding overall visual quality.
- *The proposed QGround-100K dataset.* Our uniquely compiled dataset, specifically designed to train the model on visual quality grounding in conversational contexts, enriching its ability to engage in more insightful and relevant discussions about image content and quality.

Such diverse datasets contribute to a comprehensive understanding of visual content, quality assessment, and interactive communication, making our model promising for real-world applications.

**Training objectives.** The model produces both textual outputs and segmentation masks, employing auto-regression to train the text generation component and supervised learning for the segmentation mask. In line with prior research, we apply two distinct loss functions for each output: cross-entropy loss for text generation and a hybrid of binary cross-entropy and DICE loss for mask creation. The overall loss function is represented as follows:

$$\mathcal{L} = \lambda_{txt}\mathcal{L}_{ce}(y_{txt}, \hat{y}_{txt}) + \lambda_{seg}\mathcal{L}_{seg}(y_{seg}, \hat{y}_{seg}), \qquad (6)$$

where $\hat{y}_{txt}$ is the shifted texts, $\hat{y}_{seg}$ is the ground truth mask, and $\lambda$ are loss weights. More details are given in supplementary material.

## 5 EXPERIMENTS

### 5.1 Implementation Details

*5.1.1 Training Details.* Our model is finetuned from the pretrained LLaVA-7B model [26], with CLIP-ViT-L/14-336 for visual encoding. To enhance detail capture, we follow [75] and resize the input image to $448 \times 448$. The trainable modules include the word embedding, LoRA parameters for LLM, visual projector $\phi_v$ and the mask decoder $\mathcal{D}$. We employ the AdamW [27] optimizer, setting the learning rate at 0.0003, and utilize the WarmupDecayLR scheduler, which begins with 100 warmup iterations. The batch size is set to 2 per device with 10 steps of gradient accumulation. The model is firstly pretrained with semantic segmentation datasets to obtain common semantic abilities and then finetuned with QGround-100K dataset for visual quality grounding. The total training process requires approximately 2 days on 4 NVIDIA 4090 GPUs.

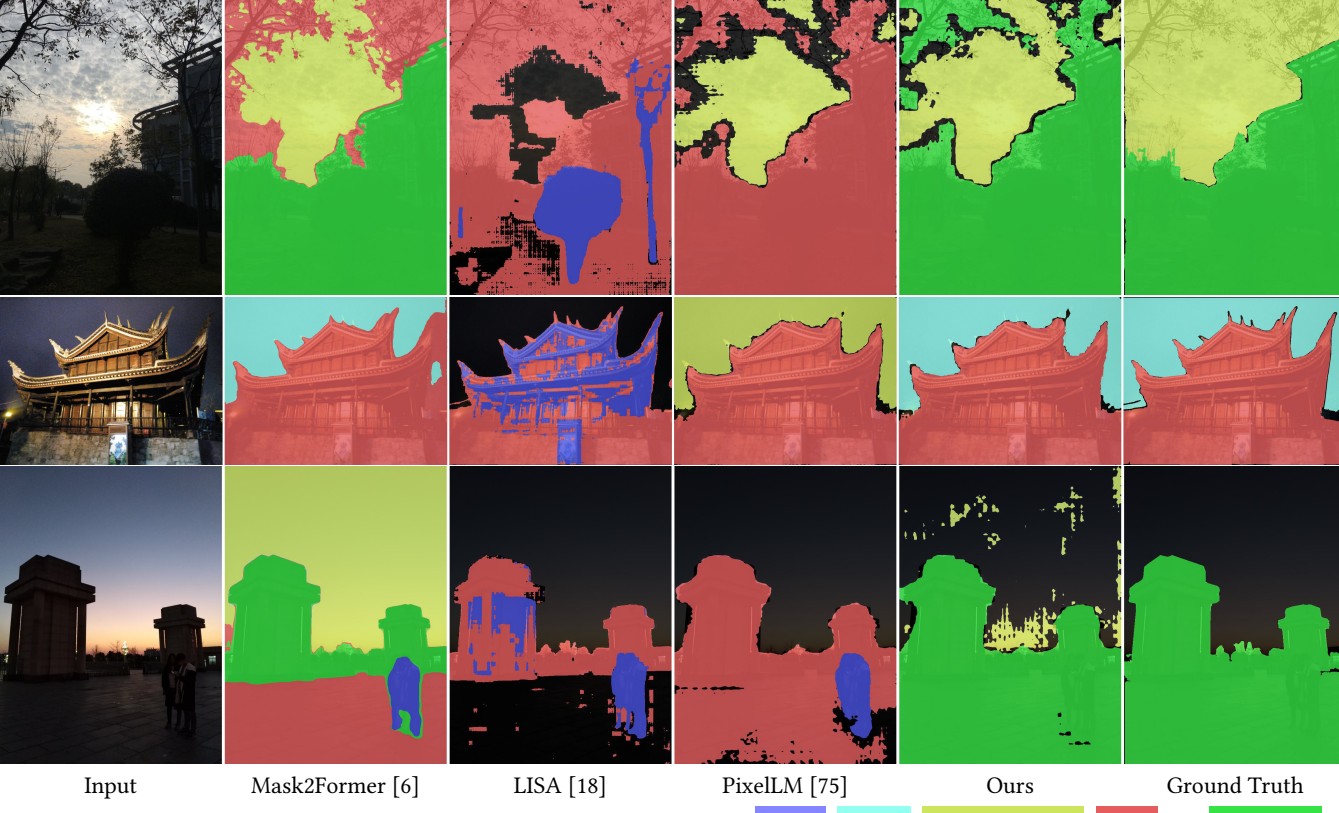

Figure 7: Visual comparison of segmentation results for distortions: **jitter** , **noise** , **overexposure** , **blur** and **low light** .

### 5.1.2 Benchmark Dataset and Evaluation Metrics.

*5.1.2 Benchmark Dataset and Evaluation Metrics.* As a new task, we establish a new benchmark for evaluating visual quality grounding. As detailed in Tab. 2, the proposed QGround-100K comprises 17, 963 unique images, each annotated with human-labeled masks. We randomly split 1, 000 as the test set. Each image is accompanied by a minimum of three distinct quality descriptions and may be associated with up to three different ground truth masks.

For quantitative evaluation, we rely on the widely recognized metrics for segmentation task, *i.e.*, the mean Intersection over Union (mIoU) and mean classification accuracy (mAcc).

## 5.2 Benchmark Performance

*5.2.1 Selected Methods and Evaluation Protocal.* Since visual quality grounding is a new task for image quality assessment, there is no existing works to compare directly as far as we know. We therefore select two kinds of methods that are closely related:

- **Semantic segmentation.** We select two exemplary segmentation techniques, *i.e.*, SegFormer [58] and Mask2Former [6], along with a recent open-vocabulary model, SAN [61], as representative methods for our analysis. Given that these models do not process textual inputs and are capable of producing only a single outcome per input image, we calculate their average performance since there are multiple ground truth masks for one input image.
- **LMM based reasoning segmentation.** This area of study is relatively new and closely aligns with our work. We choose two of the most recent contributions, LISA [18] and PixelLM [75], as methods for comparison.

Since methods based on LMMs accommodate flexible inputs and outputs, for a fair comparison, we evaluate each method using prompts like: "`<quality text>` `Please segment out distorted regions in the image.`" to obtain the corresponding mask for the identified distortions, where "`<quality text>`" is the global quality reasoning text. We use *"smaller region first"* principle to merge various segmentation masks in the event of overlaps, because it prioritizes precision and diversity in segmentation, ensuring that more details are captured and evaluated. *All these compared methods are re-trained or finetuned with QGround-100K dataset.*

*5.2.2 Results Comparison on QGround Benchmark.* According to the results shown in Fig. 7 and Tab. 3, we can notice the difference in performance between semantic segmentation models and LMM-based approaches. The traditional semantics segmentation model, especially Mask2Former, generates masks with better details and cleaner boundaries and the quantitative performance is also better. The exception, SAN, is worse likely due to its optimization for high-level segmentation tasks and lack of suitable mask decoder for visual quality grounding. The superior performance of segmentation methods is probably because they are better at the simple five-class segmentation task. Meanwhile, the LMM-based approaches face the dual challenge of identifying distortion types while concurrently generating segmentation results. Nevertheless, LLM-based methods demonstrate a significant advantage in versatility and capability over traditional segmentation techniques, offering additional abilities such as answering questions about image quality and content.

**Table 3: Quantitative comparison with segmentation methods and LMM-based methods on QGround-Test.**

| Method | jitter | | noise | | overexposure | | blur | | low light | | Average | |
|---|---|---|---|---|---|---|---|---|---|---|---|---|
| | mIoU | mAcc | mIoU | mAcc | mIoU | mAcc | mIoU | mAcc | mIoU | mAcc | mIoU | mAcc |
| SegFormer [58] | 0.327 | 0.625 | 0.136 | 0.249 | 0.264 | 0.389 | 0.515 | 0.842 | 0.274 | 0.524 | 0.373 | 0.636 |
| Mask2Former [6] | 0.401 | 0.625 | 0.089 | 0.113 | 0.223 | 0.424 | 0.566 | 0.902 | 0.290 | 0.461 | **0.403** | **0.646** |
| SAN [61] | 0.119 | 0.239 | 0.011 | 0.018 | 0.143 | 0.454 | 0.387 | 0.584 | 0.162 | 0.223 | 0.228 | 0.401 |
| LISA [18] | 0.154 | 0.688 | 0.003 | 0.003 | 0.082 | 0.102 | 0.411 | 0.682 | 0.005 | 0.006 | 0.227 | 0.436 |
| PixelLM [75] | 0.400 | 0.823 | 0.050 | 0.200 | 0.117 | 0.380 | 0.429 | 0.632 | 0.131 | 0.185 | 0.252 | 0.519 |
| Ours | 0.434 | 0.720 | 0.051 | 0.176 | 0.125 | 0.459 | 0.460 | 0.648 | 0.219 | 0.337 | **0.271** | **0.539** |

**Table 4: Ablation study of datasets used in training.**

| ID | Q & A | Seg | Q-Inst | QG-human | QG-GPT | mIoU | mAcc |
|---|---|---|---|---|---|---|---|
| I | ✔ | ✔ | | | | 0.042 | 0.113 |
| II | ✔ | ✔ | | ✔ | ✔ | 0.267 | 0.538 |
| III | | ✔ | | ✔ | ✔ | **0.275** | **0.546** |
| IV | ✔ | ✔ | ✔ | ✔ | | 0.260 | 0.531 |
| V | ✔ | ✔ | ✔ | ✔ | ✔ | 0.271 | 0.539 |

**Table 5: Ablation studies. Left: scales used in Multi-Scale Feature Abstractor; right: quality text reference in prompt.**

| Layers Used $\phi_v$ | mIoU | mAcc |
|---|---|---|
| PixelLM (23) | 0.252 | 0.519 |
| 14, 23 | 0.269 | 0.538 |
| 7, 14, 23 | **0.271** | **0.539** |

| Txt Ref | mIoU | mAcc |
|---|---|---|
| ✘ | 0.268 | 0.501 |
| ✔ | 0.271 | 0.539 |

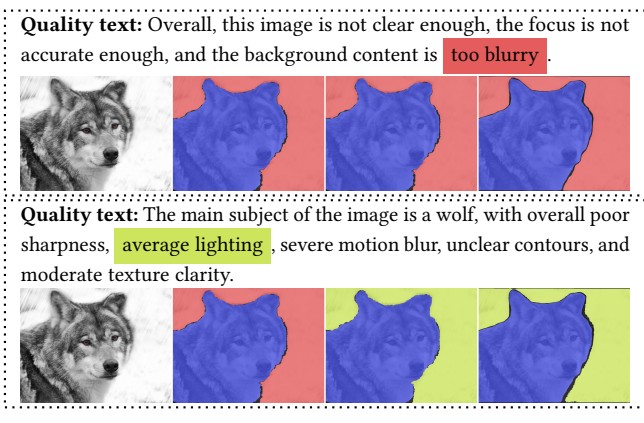

**Quality text:** Overall, this image is not clear enough, the focus is not accurate enough, and the background content is too blurry.

**Quality text:** The main subject of the image is a wolf, with overall poor sharpness, average lighting, severe motion blur, unclear contours, and moderate texture clarity.

Input Image    w/o quality text    w/ quality text    Ground Truth

**Figure 8: Examples w/ and w/o quality text in prompt.**

In LMM based approaches, PixelLM and ours outperform LISA in mask classification. This improvement is attributed to the benefit of optimizing multiple segmentation tokens, which enhances classification accuracy. On the other hand, the utilization of multi-scale features in visual projection further improves the quality concept understanding of LMM, leading to our superior performance compared with PixelLM, as illustrated in Fig. 7.

## 5.3 Analysis and Ablation Study

*5.3.1 Dataset Fusion.* We firstly examine the impact of mixed datasets training on the visual quality grounding results, as depicted in Tab. 4. Experiment I employs no quality grounding data and serves as a foundational baseline. From experiment II and III, it is observed that integrating tasks related to semantic segmentation shows little effect on the performance of quality grounding, while replacing them with Q-Instruct can produce marginally improved results. This suggests that these two tasks may be independent of one another, and their integration is feasible for developing a more capable model. When comparing IV and V, it is evident that incorporating data labeled by GPT4V is beneficial to performance. We anticipate that incorporating GPT4V will prove even more beneficial in the context of more complex data annotation processes, a potential we intend to explore in future research.

*5.3.2 Multi-scale Feature Abstractor.* Table 5 demonstrates that incorporating mid-level features significantly enhances low-level perceptual capabilities, while the inclusion of shallower level features

is also somewhat beneficial. Therefore, we empirically choose these three layers to achieve a good balance.

*5.3.3 Quality Text Reference in Prompt.* Table 5 also shows the significance of incorporating a global quality text reference. The mAcc shows a considerable improvement compared to scenarios lacking a text reference. As illustrated in Fig. 8, the model can identify distortion types mentioned in the provided text and generates corresponding results, thereby facilitating more effective interaction with users.

## 6 CONCLUSION

In this study, we pioneer the integration of visual grounding into image quality assessment, enabling a more fine-grained perception of local quality. To accomplish this objective, we collected a comprehensive dataset comprising 100K annotated samples, namely, the QGround-100K . This dataset was carefully labeled, with half of the annotations provided by human participants and the remaining half by GPT4V, thereby enhancing both the diversity and efficiency of data labeling. With this QGround-100K , we introduced a LMM-based approach that seamlessly incorporates quality grounding within multi-modal tasks. Specifically, we developed a multi-scale feature abstractor (MSFA) designed to augment the LMM's capacity to recognize low-quality attributes. Our research sets a new benchmark for the task of image quality assessment, broadening its potential applications across a wider range of fields.

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
