# OpenReview forum: "Q-Ground: Image Quality Grounding with Large Multi-modality Models"
_acmmm.org/ACMMM/2024/Conference — MM2024 Oral_

### Official Review · Reviewer_Cqap · 2024-05-09

**Rating:** 3
**Confidence:** 3

**Summary:**

This paper introduces a new framework called Q-Ground, which focuses on fine-grained visual quality grounding by combining large multimodal models with detailed visual quality analysis. The framework utilizes the QGround-100K dataset with 100,000 triples (image, quality text, distortion segmentation) for in-depth study of visual quality. The importance of perceiving the quality of detailed images, especially in local distortion, is emphasized, and a visual quality grounding task is introduced to bridge this gap. Contributions to this work include the establishment of a new benchmark for fine-grained visual quality analysis, the creation of the QGround-100K dataset, the introduction of a multi-scale visual feature abstractor based on visual quality grounding based on large multimodal models, and the establishment of a new benchmark for future research on image quality assessment

**Strengths:**

1. The Q-Ground framework introduces visual mass grounding, and the idea of combining mass reasoning with pixel-level segmentation is interesting.
2. The QGround-100K dataset is a rich resource that includes 100k triplets with diverse annotations, both from humans and automated models. It helps in comprehensive research into visual quality analysis and training models that understand visual quality at a detailed level.
3. The paper is easy to understand and the structure is clear.

**Limitations:**

1. The visual quality grounding task is inherently challenging. Unlike conventional segmentation tasks, real-world distorted images often have non-uniform local mixed distortions, so assigning only one specific distortion type to each region may not accurately reflect real-world scenarios. Thus, the information provided by the QGround-100K dataset could be limited. For example, in the middle image of Figure 7, the left side of the building shows overexposure, the surrounding area shows low light, and the building itself does not seem to have noticeable blur. However, the GT labels all of these parts as blur.
2. The QGround-100K dataset's distortion evaluation dimension is not comprehensive, covering only five common distortions. Also, based on the visual results in the paper, the distortion annotations of distorted images seem to be simplistic, only marking the foreground and background distortions, which does not align with the characteristics of local non-uniform distortions in natural images.
3. The quality descriptions provided by GPT may exhibit hallucinations. How can we ensure the reliability and credibility of GPT-assisted evaluations?
4. From the results in Table 3, the performance improvement of the proposed Q-Ground is marginal.
5. There is a lack of comparison between SAM and Semantic-SAM in terms of performance on the QGround-Test.
6. Before using Q-Ground for grounding, does the model need to evaluate the image quality first? For example, as shown in Figure 1, users need to use the prompt "Please evaluate the quality of the image" before conducting image segmentation.
7. The ablation experiments are not comprehensive. For example, it lacks ablation experiments that only use QG-human, only use QG-GPT, only use QG-human + QG-GPT, and don't use QG-human in V, etc.

**Suitability:**

2

---

### Official Review · Reviewer_JqyA · 2024-05-24

**Rating:** 4
**Confidence:** 2

**Summary:**

This paper introduces a new dataset QGround-100K which has fine-grained visual scoring annotations. Besides, it also proposes a LMM-based method that can perform both image quality answering and distortion segmentation based on text prompts.

**Strengths:**

1. The introduce of QGround-100K benchmark can help various downstream tasks such as image editing and image quality improvement.
2. The proposed model is reasonable.

**Limitations:**

1. Some typos in the paper, e.g., Fig.5
2. Could you show some error cases the GPT-4V made during annotations? and what is the error rate?

**Suitability:**

3

---

### Official Review · Reviewer_hthh · 2024-05-25

**Rating:** 4
**Confidence:** 3

**Summary:**

The paper introduces Q-Ground, a pioneering framework for pixel-level visual quality grounding that integrates large multi-modality models (LMMs) with detailed image quality analysis.  Central to this work is the QGround-100K dataset, comprising 100,000 annotated samples, which includes both human-labeled and LMM-generated annotations.

**Strengths:**

Q-Ground's core contribution is the introduction of the QGround-100K dataset, a new resource containing 100k triples (image, quality text, distorted segmentation) to facilitate in-depth research on visual quality.
The Q-Ground method not only refines the model's understanding of region-perceived image quality, but also enables it to interactively respond to complex text queries about image quality and specific distortions.

**Limitations:**

1. The paper focuses on the use of LLM for quality-related segmentation tasks, but this does not show the full advantages of LLM over traditional semantic segmentation methods, and lacks analysis in terms of open category distortion and quality assessment.
2. There is no description of the details about how the QGround-100K to obtain MOSs. There is also little analysis of quality of the dataset.
3. There is no quantitative analysis of the quality assessment ability in the experiment.

**Suitability:**

3

---

### Official Review · Reviewer_2SUJ · 2024-05-25

**Rating:** 4
**Confidence:** 4

**Summary:**

This paper introduces Q-Ground, a pioneering framework designed to deliver a comprehensive examination of local visual quality, which is crucial for detailed visual understanding. Central to this framework is the QGround-100K dataset, a novel collection comprising 100,000 triplets of (image, quality text, and distortion segmentation). This dataset is essential for in-depth investigations into visual quality and is divided into two segments: one featuring human-labeled annotations for precise quality assessment and another automatically labeled by large multimodal models (LMMs) such as GPT-4V. The latter segment enhances the robustness of model training while reducing data collection costs. Utilizing the QGround-100K dataset, the authors propose an LMM-based method equipped with multi-scale feature learning. This method enables models to perform both image quality assessment and distortion segmentation based on textual prompts.

**Strengths:**

Compared to traditional IQA methods that evaluate image quality globally, this paper introduces a pioneering framework designed to provide a comprehensive examination of local visual quality. The novelty of this work is evident, the paper is well-structured and easy to follow, and the experimental validation is thorough and convincing.

**Limitations:**

1. Regarding the human annotation phase, the paper mentions that 15 trained annotators with solid educational backgrounds are involved in the process. However, it is not explicitly stated whether each image is annotated by all 15 subjects or by a single subject per image.
2. Regarding the reliability of annotations, especially for parts of images that are fused with multiple distortions and the semantic segments with partial distortions, it is important to ensure a systematic approach to handle these complexities.
3. The annotations do not include the severity of distortions, which is important for the Image Quality Assessment (IQA) task. Additionally, the connection between partial distortions and overall quality is not clearly addressed.
4. In Table 3, it is not specified whether the average is a simple average or a weighted average according to the number of images with different distortions. Additionally, the performance gain compared to other LMM-based approaches is not clearly demonstrated. Clarifying these points would provide a better understanding of the averaging method used and the comparative effectiveness of the proposed approach.

**Suitability:**

2

---

### Meta-Review · Area_Chair_LvBD · 2024-07-07

**Recommendation:** Accept (Oral)
**Confidence:** 4

**Metareview:**

In the first round of reviews, this paper got 3 borderline acceptances and 1 borderline rejection. After the rebuttal from the authors, all reviewers updated their ratings, assigning 3 borderline acceptances and 1 weak acceptance. Given the reviews, the rebuttal, and the comments by the reviewers after the rebuttal, I recommend this paper to be accepted.